# Ultrafast-Laser-Induced Tailoring of Crystal-in-Glass Waveguides by Precision Partial Remelting

**DOI:** 10.3390/mi14040801

**Published:** 2023-03-31

**Authors:** Alexey S. Lipatiev, Sergey V. Lotarev, Tatiana O. Lipateva, Sergey S. Fedotov, Elena V. Lopatina, Vladimir N. Sigaev

**Affiliations:** Department of Glass and Glass-Ceramics, Mendeleev University of Chemical Technology, Moscow 125047, Russia

**Keywords:** direct laser writing, ultrafast laser, lanthanum borogermanate glass, channel waveguide, LaBGeO_5_, laser-induced crystallization, laser-induced vitrification

## Abstract

Space-selective laser-induced crystallization of glass enables direct femtosecond laser writing of crystal-in-glass channel waveguides having nearly single-crystal structure and consisting of functional phases with favorable nonlinear optical or electrooptical properties. They are regarded as promising components for novel integrated optical circuits. However, femtosecond-laser-written continuous crystalline tracks typically have an asymmetric and strongly elongated cross-section, which causes a multimode character of light guiding and substantial coupling losses. Here, we investigated the conditions of partial remelting of laser-written LaBGeO_5_ crystalline tracks in lanthanum borogermanate glass by the same femtosecond laser beam which had been used for their writing. Exposure to femtosecond laser pulses at 200 kHz repetition rate provided cumulative heating of the sample in the vicinity of the beam waist sufficient to provide space-selective melting of crystalline LaBGeO_5_. To form a smoother temperature field, the beam waist was moved along the helical or flat sinusoidal path along the track. The sinusoidal path was shown to be favorable for tailoring the improved cross-section of the crystalline lines by partial remelting. At optimized laser processing parameters, most of the track was vitrified, and the residual part of the crystalline cross-section had an aspect ratio of about 1:1. Thermal-induced stress emerging during the tailoring procedure was efficiently eliminated by fine post-annealing. The proposed technique suggests a new way to control the morphology of laser-written crystal-in-glass waveguides by tailoring their cross-section, which is expected to improve the mode structure of the guided light.

## 1. Introduction

The unique ability of high-precision spatially selective processing of the inside of transparent materials has made ultrafast lasers an efficient tool for the fabrication of integrated optical and photonic components and devices [1,2]. Among other modification types, the ability to locally crystallize glasses by the focused beams of various lasers drew considerable attention. This phenomenon is widely studied due to the possibility of precipitating non-centrosymmetric phases, including nonlinear optical crystals, which could be a base for functional photonic components integrated in glass [3]. Due to the nonlinear character of absorption of focused ultrashort laser pulses confined near the beam waist, femtosecond direct laser writing (DLW) is favorable for growing crystalline architectures inside transparent glasses. In particular, these architectures can be represented by continuous or, in some cases, nanoperiodical crystalline lines with a few μm sized cross-section and oriented crystalline structure formed by nonlinear optical phases such as, in particular, LaBGeO_5_ [4,5,6,7,8,9,10], LiNbO_3_ [11,12,13,14], Ba_2_TiSi_2_O_8_ [15,16], Sr_2_TiSi_2_O_8_ [17], Pb_5_Ge_3_O_11_ [18], Li_2_Ge_7_O_15_ [19]. On the contrary, spatially selective crystallization of glasses by continuous-wave [20,21,22,23] and nanosecond [24] lasers requires a certain level of absorption at the fundamental wavelength and is basically suitable for growing crystalline patterns at the glass surface though the growth of crystalline line from the surface to the inner parts of the glass volume was also demonstrated [25].

The optimization of ultrafast laser writing conditions via beam shaping and application of post-annealing enabled us to improve the homogeneity and morphology of the crystalline lines and fabricating crystal-in-glass channel waveguides formed by a virtually single-crystal structure with c-axis oriented along the waveguide [6,7,10]. However, there are still no examples of proof-of-concept integrated photonic devices and applications based on crystal-in-glass waveguides. One of the main factors limiting the feasibility of crystal-in-glass waveguides is insufficient control of the profile of crystalline lines written inside glass by ultrashort laser pulses. Temperature distribution determined by a carrot-shaped fs laser pulse absorption region in the case of DLW by the focused Gaussian beam specifies the elongated cross-section of the laser-written crystalline line, that is generally expected to increase coupling losses of the waveguide and deteriorates the mode structure of the propagating light. In the case of bilateral growth of LaBGeO_5_ crystal in lanthanum borogermanate (LBG) glass, the cross-section could be even horseshoe shaped or include two separate elongated parts [8]. Writing beam shaping by means of the liquid-crystal spatial light modulator including correction of the wavefront of the Gaussian beam [6,26] or by forming a Laguerre–Gaussian beam [27] was suggested to improve the aspect ratio of the cross-section of the LaBGeO_5_ crystalline line. A hybrid waveguide consisting of the series crystalline and amorphous sections was also reported as a way to obtain single-mode light propagation [10].

Recently, a spatially selective laser-induced vitrification method was reported [28,29]. This method facilitated writing, erasing and rewriting continuous crystal-in-glass tracks by the same femtosecond laser beam in glasses of various glass-forming systems with different crystallization ability and different precipitating phases including LaBGeO_5_, LiNbO_3_ and Ba_2_TiSi_2_O_8_ [28,29]. This improved the performance of the laser-induced crystallization method and made it reversible. Femtosecond laser-induced vitrification as a tool for 3D microfabrication was also recently applied to glass ceramics [30,31]. In transparent lithium aluminosilicate glass ceramics, femtosecond DLW induced a local decrease in the refractive index due to the vitrification of nanocrystals and facilitated the writing of Type III depressed-cladding channel waveguides [31].

In the present study, we investigated the applicability of laser-induced vitrification as a high-precision tool for tailoring the cross-section of the laser-written crystalline lines inside glass, which is expected to make it more suitable for waveguiding applications. Partial remelting of the crystalline line by the focused femtosecond beam moving in different ways relative to the crystalline line was used for this purpose.

## 2. Materials and Methods

In the experiments, we used the same LBG glass sample as in the previous studies on the laser-induced vitrification of laser-written crystalline architectures [28,29]. In fact, lanthanum borogermanate glasses allowing the precipitation of ferroelectric LaBGeO_5_ became model glasses for the investigation of direct femtosecond-laser writing of crystalline lines in glass and an object of numerous studies [4,5,6,7,8,9,10,22,24,26,27,28,29]. Trigonal ferroelectric LaBGeO_5_ is an optically positive uniaxial crystal having extraordinary (*n_e_*) and ordinary (*n_o_*) refractive indices equal to 1.8646 and 1.8247, respectively, at a wavelength of 546 nm [32]. The glass under study was fabricated by a melt-quenching technique in a Pt crucible described in more detail elsewhere [7], annealed at 640 °C for 2 h and cut into plane-parallel plates and optically polished. Its chemical composition was near 25La_2_O_3_·30B_2_O_3_·45GeO_2_ (mol%) according to X-ray fluorescence analysis.

DLW experiments were performed using FemtoLab ultrafast-laser microfabrication setup (Workshop of Photonics, Vilnius, Lithuania) based on PHAROS-SP femtosecond laser (Light Conversion, Ltd., Vilnius, Lithuania) emitting pulses at 1030 nm wavelength and on ABL1000 computer-controlled air-bearing three-axis motion control stage (Aerotech, Inc., Pittsburgh, PA, USA), providing a high-precision 3D translation of the sample relative to the focused laser beam (Figure 1). In the performed experiments, the pulse duration and pulse repetition rate were fixed as 180 fs and 200 kHz, respectively. The pulse energy indicated hereafter was measured in front of the upper surface of the sample. The laser beam was focused in the bulk of the sample within a shift of 100 µm from the glass surface by an objective lens with N.A. of 0.45, which means the actual focusing depth of about 180 µm due to the refraction of the beam as the refractive index of this glass is about 1.8 [33]. DLW was visually monitored in situ using Retiga 3000 CCD camera (Teledyne QImaging, Tucson, AZ, USA). To evaluate the share of the incident pulse energy that was absorbed by glass, the laser pulse energy of the focused laser beam transmitted through the glass was measured by Ophir PD10-C laser energy sensor as a function of the incident pulse energy. Based on this measurement and taking into account Fresnel reflection, approximate absorbed energy and radiation dose were calculated.

DLW was performed in transverse writing geometry; i.e., the glass sample plane and the direction of the translational movement of the sample were always orthogonal to the laser beam propagation direction. The laser-induced growth of a crystalline track started from a seed crystal formation performed by the stationary laser beam with gradually increasing pulse energy [7]. Once the seed crystal was formed, the tightly focused laser beam started moving along the sample at the constant depth under the glass surface, and the pulse energy and laser beam scanning speed were set to provide the growth of a continuous crystalline track.

Spatially selective vitrification experiments were performed using the same setup as DLW. Moreover, the sample was not removed from the motion stage or displaced between DLW and vitrification experiments. This allowed us to keep the sample positioning and accurate matching of the beam to the crystalline line coordinates.

The selective vitrification experiments were based on the method of 3D translation of the glass sample with a laser-written crystalline track relative to the laser beam waist along the helical path suggested in the previous study [28]. The helical translation of the sample relative to the laser beam was performed as a combination of rectilinear translation along the laser-written crystalline track (*x*-axis) at a constant velocity ***v****_x_* and harmonic oscillations along *z* and *y* axes with the same frequency *f* and a phase shift of 90° between *y*- and *z*-oscillations providing an elliptical movement in *YZ*-plane. Thus, a path of the beam waist through the sample was formed as an elliptical helix (Figure 2a). It provided a more homogeneous and smooth temperature distribution in the heated region than the straight movement of the beam. This reduces thermal stress inside the sample and the probability of microcracks appearance. Full vitrification of the laser-written crystalline track had been earlier realized by the beam moving along the helix whose axis coincided with the crystalline line to be remelted [28]. In other words, the straight path of the focal point of the laser beam used for the writing of the crystalline line in glass coincided with the axis of the helical path of the focal point of the beam used for the vitrification of this crystalline line. On the contrary, in the present experiments, the axis of the helical path of the laser beam was shifted upwards from the crystalline line (*z*-shift > 0). Besides that, the beam power was reduced to the value giving existence for a local temperature minimum at the axis of the helix below the crystal melting point. This prevented an upper part of the crystalline line from remelting. The size of the retained part of the crystalline line is determined by the size and position of the area having the temperature below the crystal melting point *T_m_* (~1200 °C for LaBGeO_5_ crystal [34]) in the central part of the helix. The area in which temperature rises above the glass transition temperature *T_g_* (672 °C for glass under study [29]) approximately corresponds to the area of the modified refractive index, and so its boundaries can be revealed using an optical microscope. A qualitative temperature distribution map in *YZ*-plane corresponding to the observed modification of the laser-exposed area is given in the lower part of Figure 2a. We also studied vitrification by the beam moving along the flat sinusoidal path oriented parallel to the glass sample plane (Figure 2b). In this case, harmonic oscillations of the sample relative to the laser beam were only along *y*-axis. Oppositely to the case of the helical path, a focal point of the beam moving along the sinusoidal path was shifted downwards from the crystalline line (*z*-shift < 0). That provided selective remelting of only the lower part of the line.

Olympus BX51 (Olympus, Tokyo, Japan) polarizing optical microscope was used for the analysis of morphology and birefringence of the laser-written crystalline lines. The side surfaces of the sample were also optically polished in order to make it possible to obtain both top and side views of the laser-written structures. A confocal polarizing micro-Raman spectrometer Ntegra SPECTRA (NT-MDT, Moscow, Russia) was applied to confirm the precipitation of LaBGeO_5_ crystalline phase in the laser-written track and to detect its preferential orientation. Excitation by the “blue” line of the argon ion laser (488 nm wavelength) was used to acquire Raman spectra.

## 3. Results

### 3.1. Direct Laser Writing of Continuous Crystalline Tracks

A series of 9 mm-long crystalline lines for further experiments on partial remelting were written based on the earlier determined DLW modes [7,10,28]. These modes provide growth of continuous homogeneous crystalline lines consisting of ferroelectric LaBGeO_5_ phase with c-axis oriented along the line and normally possessing the waveguiding ability if no defects or cracks appeared inside them during their growth. The pulse energy was 500 nJ or 520 nJ, and the scanning speed varied from 40 to 46 µm/s. These speed values are close to the maximal possible crystal growth speed at the given pulse energy and repetition rate. This condition favors unilateral crystal growth producing a single crystalline line, though in some cases, bilateral crystal growth forming a double line or a line with a horseshoe-shaped cross-section like earlier reported was observed [8]. At given focusing conditions, these exposure parameters corresponded to the absorbed radiation dose of ~0.63 MJ/cm^2^ in the case of 500 nJ pulses and 40 µm/s scanning speed. The corresponding linear density of the absorbed laser energy was ~1.55 mJ/µm. The aspect ratio of the elongated cross-section of the laser-written unilateral crystalline lines typically varied from ~6 to ~12 and its width lied in a 1.8–2.2 μm interval, while the maximum width of the surrounding glassy area with a modified refractive index achieved ~9 μm. A corresponding refractive index increase relative to the non-modified glass is ~0.006 for the crystal along *z*-axis, i.e., in the direction perpendicular to its polar axis, and <0.003 for the amorphous part of the laser-written track at a wavelength of 546 nm [28]. A typical view of the cross-section of crystalline lines in the case of bilateral or unilateral crystallization is shown in Figure 3.

The strong birefringence of LaBGeO_5_ crystals observed in the Olympus BX51 polarizing optical microscope gave clear evidence of their precipitation and allowed us to study the morphology of crystalline lines in crossed polarizers.

### 3.2. Laser-Induced Partial Remelting of the Crystalline Tracks

In the present experiments, we used the results of laser parameters optimization performed earlier for the complete vitrification of crystalline lines in LBG glass [28]. According to them, the pitch of the oscillations of the moving laser beam was set to 1 μm to provide homogeneity of the molten glass area. At the oscillation frequency *f* = 20 Hz applied in the experiments, this condition limited *v_x_* speed to 20 μm/s. The pulse energy *E_p_* varied from 300 nJ to 700 nJ. The horizontal amplitude *D_y_* of the transverse oscillations of the beam was set to 50 μm. The vertical amplitude *D_z_* varied from 24 μm to 33 μm in the case of the helical path or was equal to zero in the case of the flat sinusoidal path. Importantly, the actual amplitude of vertical oscillations of the beam waist inside the sample was larger due to refraction. To obtain its value, a nominal value of *D_z_* should be multiplied by 1.8. Thus, for instance, in the experiments represented in Figure 4a (*E_p_* = 350 nJ, *D_z_* = 33 μm) and Figure 4b (*E_p_* = 520 nJ, *D_z_* = 24 μm), the corresponding absorbed radiation dose averaged over the period of the helix was evaluated as 2.5 kJ/cm^2^ and 4.7 kJ/cm^2^, respectively. The corresponding density of the absorbed laser energy per length of the tailored track was, respectively, 2.1 mJ/µm and ~3.2 mJ/µm.

A typical modification region induced by the Gaussian beam tightly focused inside transparent material is carrot shaped (Figure 3) and has a relatively wide head zone followed by the long sharp end formed due to the filamentation of the beam [35]. Therefore, we always used the wide head zone (upper end of the modification region in the geometry of our experiment) for crystal remelting as it provides smoother temperature distribution. Due to that, a lower part of the crystalline line was exposed to remelting (Figure 2a). The vertical shift (*z*-shift) of the sinusoidal path and of the axis of the helical path of the erasing beam waist from the level at which the crystalline lines had been written varied from 4 to 9 μm. It was positive (directed to the upper surface of the sample) in the case of the helical path and negative in the case of the flat sinusoidal path.

The investigation of the remelting process and the morphology of the erased and retained crystals depending on the laser treatment parameters was carried out as a series of short passes of the erasing beam, producing partial remelting of short sections of a laser-written crystalline line. In these experiments, each partially remelted section was typically 60 μm long. Laser exposure parameters were gradually changed from section to section. Some sections were exposed under the same parameters in order to confirm the reproducibility of the obtained effects. Different combinations of *z*-shift, *D_z_* and *E_p_* facilitated the gradual variation of the erased part of the crystalline line cross-section from a few percent of its height to complete erasing (Figure 4). Some experiments included several passes of the erasing beam along the same section of the crystalline line at constant or varying *z*-shift. As a decrease in the cross-section width of the crystalline line during partial remelting of the track is almost negligible, the aspect ratio of the crystalline line cross-section changes proportional to its height. Boundaries of the region with the modified refractive index surrounding the partially erased line indicate the area heated above the glass transition temperature by the erasing laser beam.

Raman mapping was used to confirm the morphology and crystalline nature of the retained part of the laser-written line after its partial erasing (Figure 5). The integral intensity of Raman scattering in the wavenumber range of 385–400 nm was mapped because this range includes a characteristic peak of LaBGeO_5_ crystal with a maximum at 395 cm^−1^ [28]. The image area corresponding to the crystal completely coincided with the cross-polarized optical image and the Raman map.

A main drawback of partial erasing of crystalline lines using the helical path is a sort of ripple often appearing in the retained part of the crystalline lines. It is expressed in periodical brightness oscillation in crossed polarizers, which indicates a periodical thickness decrease in the retained crystalline part, probably up to its full periodical vitrification (Figure 6a,b). Its period coincided with the pitch of the helical path of the erasing beam equal to 1 μm. This effect is caused by the sharp bottom end of the carrot-shaped heating area of the focused femtosecond laser beam when it moves in the top sector of the helix and passes above the crystalline line. Periodical variation of the refractive index can also be observed in modified glassy regions in the bottom part of the heating area (Figure 6c). Thus, the sharp and strong temperature peak in the bottom of the “carrot” induces significant alternation of heating conditions. It means that the pitch of the helix equal to 1 μm in the performed experiments occurred to be too large to provide overlapping of the temperature peaks in the adjacent turns of the helix. At the same time, the bottom of the “carrot” occurred to be long enough to affect the upper part of the crystalline line at the given helix radius and position. As a matter of fact, the revealed effect can be applied for high-precision tailoring crystalline lines with periodically varying width or periodical glass/crystal lines, but in the case of the crystal-in-glass waveguides, it is considered to be detrimental.

This problem can be solved by increasing the vertical size *D_z_* of the helix with an additional shift upwards. Anyway, the quality of the retained crystalline line becomes less predictable because the actual length of the “carrot” bottom affecting the crystal also depends on the pulse energy. To completely avoid this drawback, a flat sinusoidal path of the erasing beam (Figure 2b) was applied instead of the helical one. In this case, *D_z_* = 0 and the carrot-shaped heating area of the laser beam is always below the crystalline line. Thus, remelting of the crystal is realized only by its wide upper part. Examples of top, side and crosscut views of the erased tracks are shown in Figure 7. The height of the remelting area could be precisely controlled by setting a combination of *z*-shift and pulse energy because increasing the pulse energy expanded the remelting area. Pulses of 520 nJ were applied in most of the further experiments. The corresponding density of the absorbed laser energy per length of the tailored track was ~3.2 mJ/µm, and the absorbed radiation dose averaged over the period of the sinusoidal path was 6.8 kJ/cm^2^. Partial erasing in this way enabled the tailoring of the crystalline lines with a nearly 1:1 aspect ratio of the cross-section (Figure 7b,d).

After tuning the laser treatment conditions, partial vitrification along the sinusoidal path was reproduced at the full length of the tracks, i.e., 9 mm-long crystalline lines with improved aspect ratio were tailored (Figure 7d–h). A series of successfully tailored crystalline lines confirmed the high precision and reproducibility of this method. Sometimes erasing along the sinusoidal pass caused the emergence of microcrack in surrounding glass along the line due to the larger thermal stress (Figure 7d), but this crack does not involve the retained crystal. The sample with a 9 mm-long laser-tailored crystalline line was polished at two opposite sides in order to expose the facets of the line to the side surfaces of the sample. However, this operation caused periodical cracking of the line throughout its length (Figure 8) due to the thermal stress induced in the laser-modified region during the partial erasing of the crystalline line. Fine annealing of the LBG glass sample with the laser-tailored tracks was then performed according to the earlier developed regime [10], i.e., heat treatment at 640 °C (which is nearly 30 °C below *T_g_*) for 4 h and slow cooling at a rate of 25 C° h^−1^ to room temperature. Examination of the post-annealed sample in crossed polarizers showed that stress-induced birefringence in glass surrounding the crystalline lines was completely avoided, but the laser-induced refractive index modification of glass, which had appeared during partial erasing of the lines, could still be observed.

## 4. Discussion

The obtained results suggest a novel method to control the morphology of laser-written crystalline lines inside glass by means of their accurate partial remelting. Unlike in the earlier reported case of full erasing of the crystalline tracks [28], the helical path of the erasing beam is shown to be not optimal for partial erasing because of the detrimental effects of the sharp bottom end of the heating area of the focused laser beam on the retained part of the crystal. Instead, a flat sinusoidal path of the erasing beam provides finely controlled tailoring of the crystalline line with the desired aspect ratio. On the other hand, the sinusoidal path produces less smooth temperature distribution along the beam, which can cause the occasional emergence of microcracks parallel to the glass surface along the track. However, in most cases, this crack was far enough from the crystalline track and will not affect its performance as a waveguide. The thermally induced stress can be released by post-annealing, which is strongly recommended before any further operations with the laser-tailored crystalline line. Importantly, post-annealing was also reported to improve the homogeneity of the crystalline structure of continuous crystalline lines laser-written in the inside of glass by the example of LaBGeO_5_ [10] and Ba_2_TiSi_2_O_8_ [16] crystals.

The obtained crystalline lines are expected to be suitable as single-mode crystal-in-glass waveguides though a reduced size of the input facet will complicate light coupling. Moreover, an increased refractive index in the surrounding amorphous area modified by the laser beam also imparts waveguiding properties to it [10]. Therefore, a certain part of incoming light can be guided by the amorphous region outside the crystalline line. In order to provide the feasibility of the laser-tailored integrated crystal-in-glass LaBGeO_5_ waveguides, a problem of accurate coupling has to be solved in order to efficiently confine the input light, e.g., from a standard single-mode fiber to the facet of the crystalline waveguide.

The significant technical limitation of this method is the necessity of high-precision positioning of the sample during laser tailoring. The best results can be achieved when writing and partial erasing of the crystalline line are performed as stages of one processing cycle within which the glass sample remains in the same position.

Thus, the proposed method can be applied to improve the performance of the laser-written crystal-in-glass waveguides by tailoring their cross-section, which is expected to improve the mode structure of the guided light up to obtaining single-mode channel waveguides. Though it is shown here only by the example of LBG glass, it is expected to be versatile and suitable for tailoring the cross-section of the crystalline lines consisting of other phases precipitated in glasses of other glass-forming systems. This is proved by its principal similarity to the method of complete laser-induced vitrification of the crystalline tracks, whose feasibility was demonstrated for different glasses with significantly varying crystallization properties.

## Figures and Tables

**Figure 1 micromachines-14-00801-f001:**
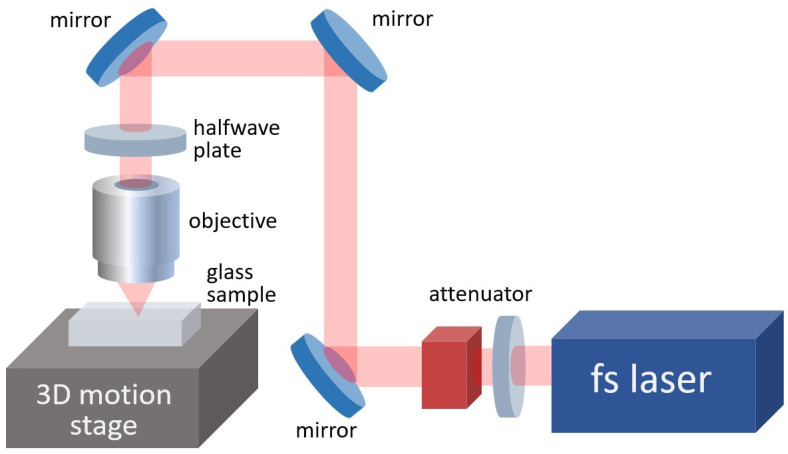
Simplified layout of the laser microfabrication setup.

**Figure 2 micromachines-14-00801-f002:**
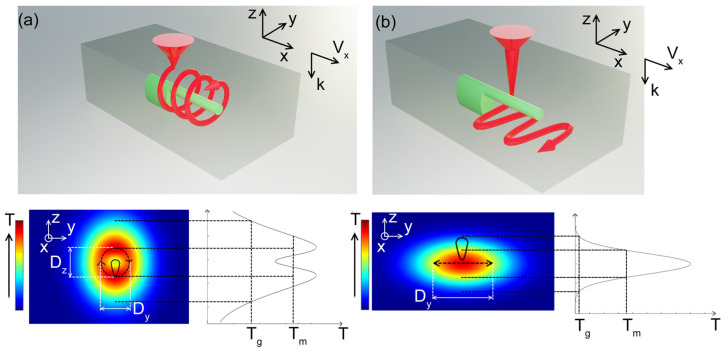
Scheme of laser-induced vitrification of the crystalline lines by the beam waist moving translationally along the helical (**a**) or flat sinusoidal (**b**) path relative to the glass sample and the corresponding qualitative temperature distribution maps in the cross-section plane. *k* and *v_x_* indicate the erasing laser beam propagation direction and the beam translation direction along *x*-axis relative to the sample, respectively. *T*, *T_g_* and *T_m_* are for temperature, glass transition temperature and crystal melting point, respectively.

**Figure 3 micromachines-14-00801-f003:**
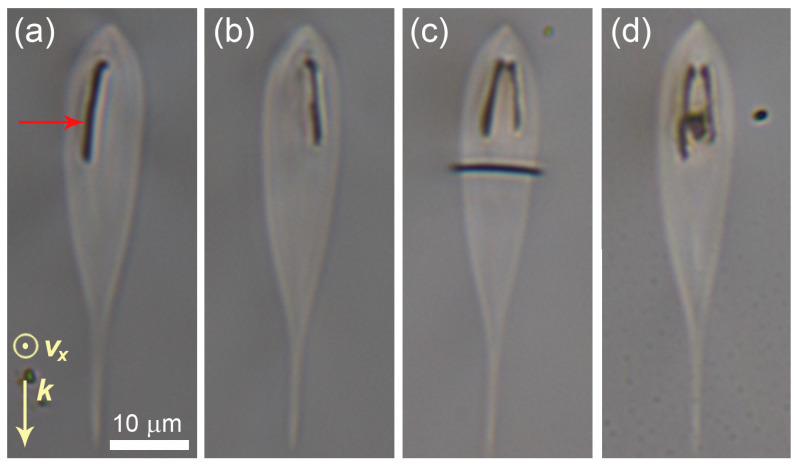
Brightfield optical images of the cross-sections of the laser-written tracks containing LaBGeO_5_ crystalline lines in the case of unilateral (**a**,**b**) and bilateral (**c**,**d**) crystal growth. Crystal is shown by the red arrow. *k* and *v_x_* indicate the writing laser beam propagation direction and the sample translation direction, respectively. All the images have the same scale.

**Figure 4 micromachines-14-00801-f004:**
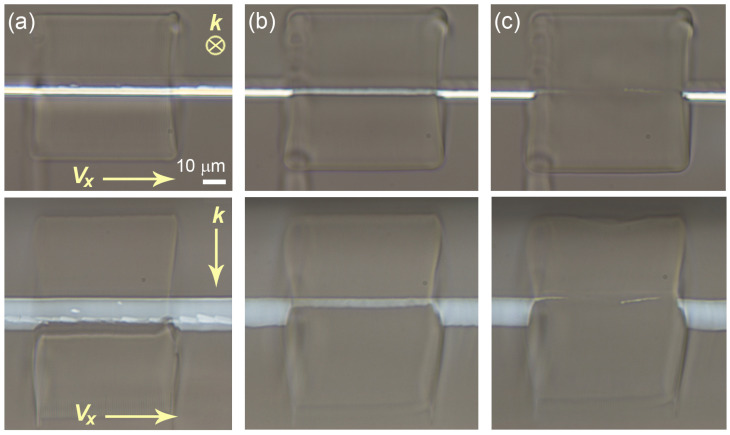
Cross-polarized optical images of the top (upper row) and side (lower row) views sections of a LaBGeO_5_ crystalline line erased by ~25% (**a**) (*E_p_* = 350 nJ, *D_z_* = 33 μm), ~80% (**b**) (*E_p_* = 520 nJ, *D_z_* = 24 μm), and 100% (**c**) (*E_p_* = 520 nJ, *D_z_* = 21 μm) of its height by the laser beam moving along the helical path (*z*-shift is +8 μm in all the cases). Vectors *k* and *v_x_* indicate the erasing laser beam propagation direction and the beam translation direction along *x*-axis relative to the sample, respectively. All images have the same scale.

**Figure 5 micromachines-14-00801-f005:**
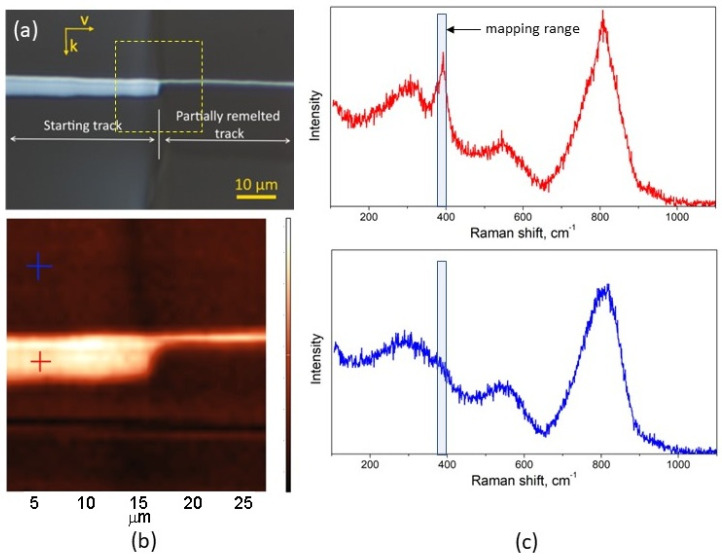
Cross-polarized optical image of the side view of a LaBGeO_5_ crystalline line partially erased the beam with *E_p_* = 350 nJ, *D_z_* = 33 μm (**a**); Raman map (**b**) of the integrated Raman scattering intensity in the wavenumber range 385–400 cm^−1^ recorded in the area indicated by the yellow dashed square (**b**); Raman spectra of glass and the crystalline line are acquired in the points addressed by the blue and red crosses, respectively (**c**). Vectors *k* and *v_x_* indicate the erasing laser beam propagation direction and the beam translation direction along *x*-axis relative to the sample, respectively. All images have the same scale.

**Figure 6 micromachines-14-00801-f006:**
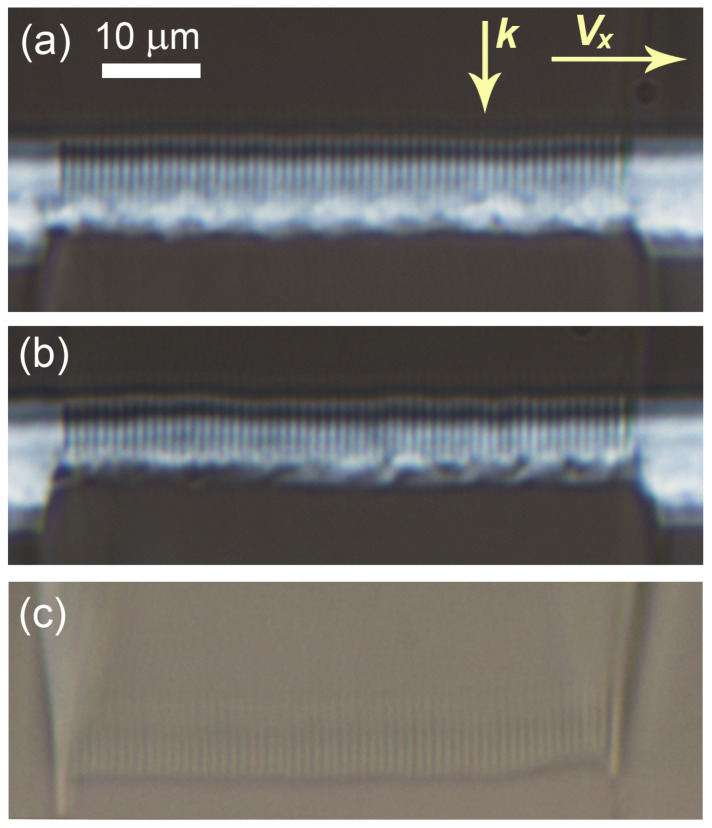
Cross-polarized optical images of the side view of the retained crystalline lines with periodically varying width in LBG glass (**a**,**b**) and the periodical variation of the refractive index of glass in the bottom of the helix (**c**) caused by the erasing beam moving along the helical path. The erasing conditions are: *E_p_* = 350 nJ, *D_z_* = 30 μm, *z*-shift = +8 μm, +12 μm in two consecutive passes (**a**), *z*-shift = +8 μm, +12 μm, +14 μm in three consecutive passes (**b**), *z*-shift = +8 μm (**c**). *k* and ***v_x_*** indicate the erasing laser beam propagation direction and the beam translation direction along *x*-axis relative to the sample, respectively. All the images have the same scale.

**Figure 7 micromachines-14-00801-f007:**
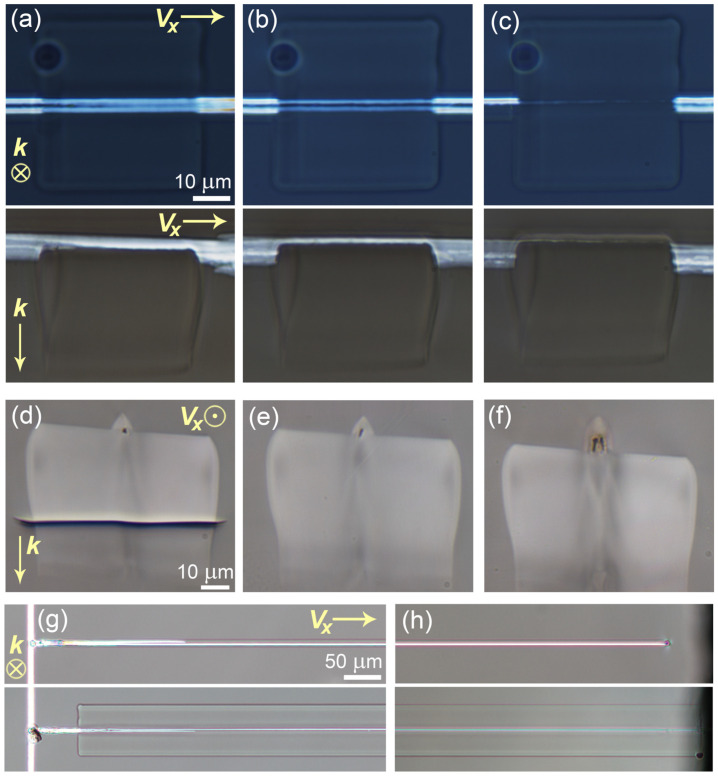
Optical images of the laser-written LaBGeO_5_ crystalline lines in LBG glass partially erased by the femtosecond laser beam using the flat sinusoidal path: top and side views (cross-polarized) of the line sections erased by 60% (**a**), 85% (**b**) and 100% (**c**) of their cross-section height (*Ep* = 520 nJ, *z*-shift = −5 μm, −3 μm, −1 μm, respectively); and the cross-cut view of long partially erased unilateral (**d**,**e**) and bilateral (**f**) crystalline lines; top view (cross-polarized) of the beginning (**g**) and the end (**h**) of 9 mm-long starting (top) and laser-tailored (bottom) crystalline lines. *k* and *v_x_* indicate the erasing laser beam propagation direction and the beam translation direction along *x*-axis relative to the sample, respectively.

**Figure 8 micromachines-14-00801-f008:**
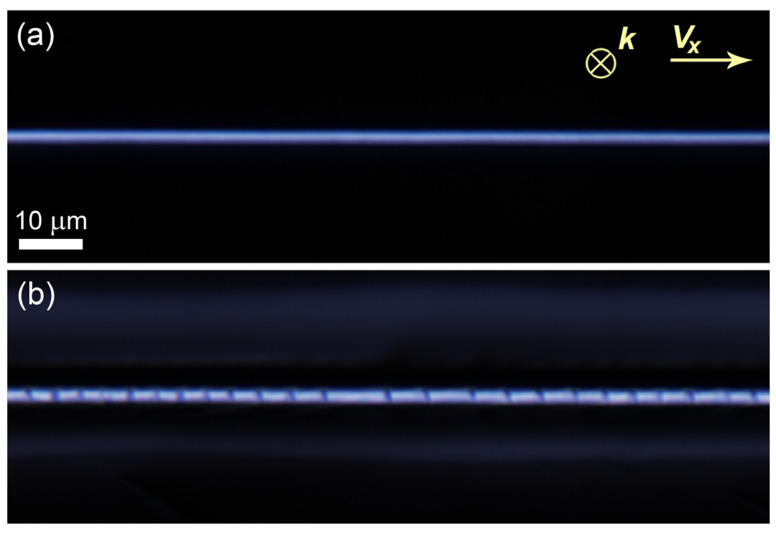
Cross-polarized optical images of the top view of the laser-tailored LaBGeO_5_ crystalline line in LBG glass before (**a**) and after (**b**) exposing its facets to the side surfaces of the sample by polishing.

## Data Availability

Not applicable.

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
