# Peer review of "Ultrafast-Laser-Induced Tailoring of Crystal-in-Glass Waveguides by Precision Partial Remelting"

_micromachines, 2023, doi:10.3390/mi14040801_

Round 1

Reviewer 1 Report

Please address the following comments in order to improve the manuscript quality

·        Line 131-132 – ‘reduced temperature below 131 the crystal melting point Tm in the central part of the helix.’ Please provide crystal melting point Tm and glass transition temperature Tg of the material.

·         Figure 2 (lines 140-142) – please provide more details regarding calculation of temperature distribution maps. Heat conductance parameters for the material are also welcomed.

·         Line 153-154 – please provide refractive index of LaBGeO5.

·         Line 156-157 ‘The pulse energy was 500 nJ or 520 nJ and the scanning speed was varied from 40 to 46 μm/s.’ and Line 179-181 ‘At the oscillation frequency f = 20 Hz applied in the experiments, this condition limited vx speed to 20 μm/s. The pulse energy Ep was varied from 300 nJ to 700 nJ.’ Such parameters do not allow to reproduce the experiments correctly. Please, evaluate the radiation dose absorbed in material both for crystallization of LaBGeO5 and for re-melting into glass phase.

·         Line 162-164 ‘The aspect ratio of the elongated cross-section of the laser-written unilateral crystalline lines 162 typically varied from ~6 to ~12 and its width lied in 1.8-2.2 μm interval, while the maximum width of the surrounding glassy area with a modified refractive index achieved ~9 μm.’ Please provide refractive index change in glass under irradiation.

·         Line 177 – reference seems to be incorrect

·         Line 186-188 ‘Since a typical nonlinear absorption region of the Gaussian beam tightly focused inside transparent material is carrot-shaped, the sharp end of the “carrot” being directed towards the focal point, we used the opposite, thicker end for crystal remelting as it provides smoother temperature distribution.’ Such shape of laser-affected zone arises from noninear interaction of propagating pulse with the medium and is known as beam filamentation (see /10.1016/j.physrep.2006.12.005 for details). Please elaborate for a more accurate discussion regarding the shape of laser-affected zone.

Author Response

Response to Reviewers

We are very grateful to Editor and Reviewers for taking the time to consider our manuscript and giving helpful recommendations. Our response is given below, and all modifications in the revised manuscript are tracked by the blue font.

Reviewer #1: Please address the following comments in order to improve the manuscript quality

  • Line 131-132 – ‘reduced temperature below 131 the crystal melting point Tm in the central part of the helix.’ Please provide crystal melting point Tm and glass transition temperature Tg of the material.

Tm and Tg values are added to the text.

  • Figure 2 (lines 140-142) – please provide more details regarding calculation of temperature distribution maps. Heat conductance parameters for the material are also welcomed.

The temperature distribution maps are just indicative illustrations in order to give a more vivid presentation of the suggested method and to facilitate its perception rather than the calculated temperature field. Therefore there is no quantitative scale for the used color grade. The maps are based on the observed morphology of the laser-induced modifications. A corresponding comment is added to the text. Unfortunately, the heat conductance parameters of LBG glass haven’t yet been reported.

  • Line 153-154 – please provide refractive index of LaBGeO5.

We added the refractive index of LaBGeO5 crystal and a corresponding reference.

  • Line 156-157 ‘The pulse energy was 500 nJ or 520 nJ and the scanning speed was varied from 40 to 46 μm/s.’ and Line 179-181 ‘At the oscillation frequency f = 20 Hz applied in the experiments, this condition limited vx speed to 20 μm/s. The pulse energy Ep was varied from 300 nJ to 700 nJ.’ Such parameters do not allow to reproduce the experiments correctly. Please, evaluate the radiation dose absorbed in material both for crystallization of LaBGeO5 and for re-melting into glass phase.

We added the evaluation of the absorbed radiation dose. As several exposure parameters varied, it’s impossible to give one relevant range of the absorbed radiation dose, so we gave a few examples of its value for some applied combinations of exposure parameters. For instance, in the case of crystalline line growth at given focusing conditions, 500 nJ pulses corresponded to 1,03 MJ/cm2 incident radiation doze in the beam waist and nearly 0,63 MJ/cm2 absorbed radiation doze. Formally, the absorbed radiation dose during remelting of the crystalline track is two orders of magnitude smaller than during writing because the helical or sinusoidal path is respectively longer than the straight path, while the speed of translation along the track (x-axis translation) has the same order of magnitude. Thus, to provide a better understanding of energy parameters, we also added the linear density of the absorbed energy (energy per unit of the track length), which is about 1,5 mJ/um during crystal writing and about 2-3 mJ/um in different regimes of its partial remelting.

  • Line 162-164 ‘The aspect ratio of the elongated cross-section of the laser-written unilateral crystalline lines 162 typically varied from ~6 to ~12 and its width lied in 1.8-2.2 μm interval, while the maximum width of the surrounding glassy area with a modified refractive index achieved ~9 μm.’ Please provide refractive index change in glass under irradiation.

A refractive index increase relative to the non-modified glass is ~0.006 for the crystal and <0.003 for the amorphous part of the laser-written track at the wavelength of 546 nm (as reported in Ref. 28). This information is added to the manuscript.

  • Line 177 – reference seems to be incorrect

The reference is corrected.

  • Line 186-188 ‘Since a typical nonlinear absorption region of the Gaussian beam tightly focused inside transparent material is carrot-shaped, the sharp end of the “carrot” being directed towards the focal point, we used the opposite, thicker end for crystal remelting as it provides smoother temperature distribution.’ Such shape of laser-affected zone arises from noninear interaction of propagating pulse with the medium and is known as beam filamentation (see /10.1016/j.physrep.2006.12.005 for details). Please elaborate for a more accurate discussion regarding the shape of laser-affected zone.

Thank you for the recommended paper. We revised the discussion regarding the shape of the laser-affected zone.

Reviewer 2 Report

In this paper the authors present a method to tailor the waveguides fabricated by ultrafast laser crystallization in glass. I see the interest of the technique, as I understand that the shape of the crystalized glass is not ideal and some post processing is needed. I regret to say that after reading the text I couldn’t understand what the authors approach to the problem is exactly.

The article is very difficult to follow, the figures, in my opinion, are not clear, and I couldn’t find the explanation in the text. I recommend a deeply revision of the manuscript. Moreover, the English is not correct at some points, the sentences are too long, and I think this is one of the reasons why the manuscript is so difficult to understand.

There are also some typing errors that shows the little effort made by the authors prior to sending the final version. Take for example:

Line 35

“… considerable attention and has been being widely studied due to the possibility to …”

On this respect I can point some other issues, for example:

The acronym DLW appears in line 112 for the first time and without being properly defined previously, I understand they refer to Direct Laser Writing.

On a more practical aspect, I don’t see how the authors manage to make what they say in Line 95. My question is, where are they positioning the energy meter? After the microscope lens? Isn’t there the fluence enough to damage the sensor? What I see more likely is that they have measured before the objective and then applied its transmittance, that can be measured in advance with a less intense source.

Line 95:

“The pulse energy indicated hereafter was measured in 95front of the upper surface of the sample”.

Regarding the figures,

For example, Figure 1, I don’t see it necessary, it is very big, and it does not give new information, I think the writing procedure is something that is clear in the text.

As I mention, they are not clear to me, maybe they have the best resolution the authors can achieve, I understand that, but just an indication of what is displayed would help the reader to identify the important parts. I am sorry but I didn’t know what I was supposed to see in most of them.

In my opinion the paper is not suitable for publication in Micromachines in its present form, I recommend rejection.

Author Response

Response to Reviewers

We are very grateful to Editor and Reviewers for taking the time to consider our manuscript and giving helpful recommendations. Our response is given below, and all modifications in the revised manuscript are tracked by the blue font.

Reviewer #2: In this paper the authors present a method to tailor the waveguides fabricated by ultrafast laser crystallization in glass. I see the interest of the technique, as I understand that the shape of the crystalized glass is not ideal and some post processing is needed. I regret to say that after reading the text I couldn’t understand what the authors approach to the problem is exactly.

The article is very difficult to follow, the figures, in my opinion, are not clear, and I couldn’t find the explanation in the text. I recommend a deeply revision of the manuscript. Moreover, the English is not correct at some points, the sentences are too long, and I think this is one of the reasons why the manuscript is so difficult to understand.

We thank the reviewer for the thorough consideration of the manuscript. We revised the text trying to improve its language and split the excessively long sentences. Some graphical and textual explanations are added to the figures to make them clear.

There are also some typing errors that shows the little effort made by the authors prior to sending the final version. Take for example:

Line 35

“… considerable attention and has been being widely studied due to the possibility to …”

This phrase is revised. The text is checked for typos and spelling mistakes.

On this respect I can point some other issues, for example:

The acronym DLW appears in line 112 for the first time and without being properly defined previously, I understand they refer to Direct Laser Writing.

The definition is added.

On a more practical aspect, I don’t see how the authors manage to make what they say in Line 95. My question is, where are they positioning the energy meter? After the microscope lens? Isn’t there the fluence enough to damage the sensor? What I see more likely is that they have measured before the objective and then applied its transmittance, that can be measured in advance with a less intense source.

Line 95:

“The pulse energy indicated hereafter was measured in 95front of the upper surface of the sample”.

To measure the power after the objective lens, the power meter sensor was placed after the objective lens instead of the glass sample. It was placed out of the focus plane of the objective lens, so the beam diameter was large and the corresponding beam intensity small enough to keep the sensor safe.

The real-time control of the beam power during the experiment was performed by measuring the power of the small part of the laser light reflected by the glassy plate introduced into the path of the beam before the objective.

As the ratio between the power value after the objective lens and that of the reflected part of the beam was known, the real-time power after the objective lens could be easily calculated.

Regarding the figures,

For example, Figure 1, I don’t see it necessary, it is very big, and it does not give new information, I think the writing procedure is something that is clear in the text.

As in previous papers, we often received reviewer’s requests to add the setup layout if it was absent, we decided that it could be helpful for readers who are not familiar with laser microfabrication. We suggest that it should remain in the paper

As I mention, they are not clear to me, maybe they have the best resolution the authors can achieve, I understand that, but just an indication of what is displayed would help the reader to identify the important parts. I am sorry but I didn’t know what I was supposed to see in most of them.

Unfortunately, the quality of the microscope images is mainly determined by the optical resolution of the microscope and cannot be improved by increasing a number of pixels forming the image. Their quality is typical for optical analysis of such structures, while a better resolution is achieved using electron microscopy. However, we consider it sufficient to express the main features of the microstructures which are discussed in the paper. To make the pictures more intelligible, we added some graphical explanations to the images.

Reviewer 3 Report

This work demonstrates a novel method to control the morphology of laser-written crystalline lines inside glass by means of their accurate partial remelting. The authors come up a new flat sinusoidal path of the erasing beam provides controlled tailoring of the crystalline line with the desired aspect ratio, which can be potentially applied as single-mode crystal-in-glass waveguides. The authors conduct systematic studies of the pros and cons of this new method and propose measures to eliminate the impact of disadvantages. The periodical thickness decrease in the retained crystalline in the helical path can be avoided by applying this new flat sinusoidal path. Although additional microcracks in the flat sinusoidal path cause extra concerns, this drawback does not impact the its performance as a waveguide and the extra thermal stress can be released by post-annealing. The method this work introduces is novel and significant. The results are systematic and solid. Thus, I recommend publishing this manuscript as it is.

Author Response

Response to Reviewers

We are very grateful to Editor and Reviewers for taking the time to consider our manuscript and giving helpful recommendations. Our response is given below, and all modifications in the revised manuscript are tracked by the blue font.

Reviewer #3: This work demonstrates a novel method to control the morphology of laser-written crystalline lines inside glass by means of their accurate partial remelting. The authors come up a new flat sinusoidal path of the erasing beam provides controlled tailoring of the crystalline line with the desired aspect ratio, which can be potentially applied as single-mode crystal-in-glass waveguides. The authors conduct systematic studies of the pros and cons of this new method and propose measures to eliminate the impact of disadvantages. The periodical thickness decrease in the retained crystalline in the helical path can be avoided by applying this new flat sinusoidal path. Although additional microcracks in the flat sinusoidal path cause extra concerns, this drawback does not impact the its performance as a waveguide and the extra thermal stress can be released by post-annealing. The method this work introduces is novel and significant. The results are systematic and solid. Thus, I recommend publishing this manuscript as it is.

We thank the reviewer for taking the time to read and consider the manuscript and for appreciation of our study.

Reviewer 4 Report

This paper presents a study on the ultrafast-laser induced tailoring of crystal-in-glass waveguides by precision partial remelting. The paper was not well organized, and very limited results were presented. Additionally, the authors did not carry out a rigorous literature survey, which did not summarize the key advances addressed in the field. The novelty of the work is seriously lacking compared with the previous publications. Datils regarding how to carry out the ultrafast-laser-induced tailoring of crystal-in-glass waveguides are missing. There are no repetitions of the tests as well as the measurements, making the obtained results unconvincing.

Author Response

Response to Reviewers

We are very grateful to Editor and Reviewers for taking the time to consider our manuscript and giving helpful recommendations. Our response is given below, and all modifications in the revised manuscript are tracked by the blue font.

Reviewer #4: This paper presents a study on the ultrafast-laser induced tailoring of crystal-in-glass waveguides by precision partial remelting. The paper was not well organized, and very limited results were presented.

We thank the reviewer for the critical perusal of our paper. We tried to improve the organization of the text in the revised version. In the manuscript, we report a novel technique of ultrafast-laser microfabrication by the example of tailoring crystalline lines in LBG glass as a model object and experimental results, which we considered adequate to give a complete and aggregate picture of its application.

Additionally, the authors did not carry out a rigorous literature survey, which did not summarize the key advances addressed in the field. The novelty of the work is seriously lacking compared with the previous publications.

We can’t agree with this remark. Our group has been conducting research in the field of laser-induced crystallization of glasses for 15 years, and to the best of our knowledge, there are no articles concerning the problem of controlled partial vitrification of the laser-written crystalline architectures in oxide glasses to compare our results with, except those already mentioned in the study. Unfortunately, the reviewer didn’t indicate any important previous publications and the key advances closely related to our study, which he meant as missing. In the revised version, we added two references to give a view of laser-induced vitrification in a broader context.

Datils regarding how to carry out the ultrafast-laser-induced tailoring of crystal-in-glass waveguides are missing. There are no repetitions of the tests as well as the measurements, making the obtained results unconvincing..

During this study, numerous repetitions of the laser-induced partial amorphization of the crystalline lines allowed us to deduce the laser exposure parameters for precise laser tailoring. We didn’t mention this in the text explicitly because, normally, it is meant by default for experimental data suggested for publishing. In the revised version, we added some words concerning repetitions.

Unfortunately, the reviewer didn’t specify what kind of measurements would make the obtained results convincing, in his opinion. We believe that optical polarized light microscopy data together with our earlier published data on the structure and morphology of the crystalline lines in LBG glass, which we referred to in the manuscript [Refs. 7,10,28,29] provides a consistent description of the effect under study sufficient for its reproduction and usage. However, we also added Raman mapping data confirming the performed partial vitrification of the crystalline line.

Round 2

Reviewer 4 Report

ACCEPT.